# Characterising the patient experience of diagnostic lumbar puncture in idiopathic intracranial hypertension: a cross-sectional online survey

William J Scotton,[1,2] Susan P Mollan,[1,3] Thomas Walters,[1] Sandra Doughty,[4] Hannah Botfield,[1,5] Keira Markey,[1,2] Andreas Yiangou,[1,2] Shelley Williamson,[4] Alexandra J Sinclair[1,5]

SW and AJS are joint senior authors.

For numbered affiliations see end of article.

**Correspondence to**
Dr Alexandra J Sinclair;
a.b.sinclair@bham.ac.uk

## ABSTRACT

**Objectives** Patients with idiopathic intracranial hypertension (IIH) usually require multiple lumbar punctures (LPs) during the course of their disease, and often report significant morbidity associated with the procedure. The aim of this study was to assess the patient's experience of diagnostic LP in IIH.

**Design, methods and participants** A cross-sectional study of patients with IIH was conducted using an anonymous online survey, with the questions designed in collaboration with IIH UK (the UK IIH charity). Responses were collated over a 2-month period from April to May 2015. Patients were asked to quantify responses using a Verbal Rating Score (VRS) 0–10 with 0 being the minimum and 10 the maximum score.

**Results** 502 patients responded to the survey, of which 463 were analysed for this study. 40% of patients described severe pain during the LP (VRS ≥8), and the median pain score during the LP was 7 (VRS, IQR 5–7). The majority of patients felt they received insufficient pain relief (85%). Levels of anxiety about future LPs were high (median VRS 7, IQR 4–10), with 47% being extremely anxious (VRS ≥8). LPs performed as an emergency were associated with significantly greater pain scores compared with elective procedures (median 7, IQR 5–7 vs 6, IQR 4–8, p=0.012). 10.7% went on to have an X-ray-guided procedure due to failure of the initial LP, and the body mass index was significantly higher in this group (mean kg/m$^2$ 40.3 vs 35.5, p=0.001). Higher LP pain scores (VRS) were significantly associated with poorly informed patients (Spearman's correlation, r=−0.32, p<0.001). Patients felt more informed when the LP was performed by a specialist registrar compared with a junior doctor (median 7 vs 5, p=0.001) or a consultant compared with a junior doctor (median 8 vs 5, p<0.001).

**Conclusions** This study was commissioned by the IIH patient group and is the first to document the patient experience of diagnostic LPs in IIH. It shows that the majority of these patients are experiencing significant morbidity from pain and anxiety. Patient experience of LP may be improved through changing clinical practice to include universal detailed preprocedural information, and where possible, avoiding emergency LPs in favour of LPs booked on an elective day-case unit.

### Strengths and limitations of this study

► This large sample size UK survey is the first known to directly and specifically document the patient experience of diagnostic lumbar punctures in idiopathic intracranial hypertension, and it confirms that a significant number of these patients are experiencing morbidity from pain and anxiety related to the procedure.

► The use of an online questionnaire ensured anonymity, thus increasing the likelihood of honest reporting by patients of their subjective experiences of the procedure.

► Given the self-report nature of this study, the results may be susceptible to recall bias, thus limiting the generalisability of our findings.

## INTRODUCTION

Idiopathic intracranial hypertension (IIH) is characterised by raised intracranial pressure which can cause papilloedema with significant visual loss in some, as well as severe disabling headaches which significantly impact on quality of life in the majority.[1 2] The diagnostic criteria for IIH is based on an elevated lumbar puncture (LP) opening pressure (≥250 mm CSF in adults) in a properly performed LP.[3]

Many patients have multiple LPs during the disease course typically to assess disease severity, and in some cases as a therapeutic strategy. Established complications of LPs include local discomfort, low-pressure headaches and more rarely infection or local haemorrhage.[4] We have been made aware of an additional significant complication of LPs voiced by the patients themselves. The patients describe a very negative experience of LPs associated with anxiety, fear and pain during and after the procedure. The National charity IIH UK (Registered Charity in England and

Wales no 1143522 and Scotland SCO43294) approached us with concerns about the patients with IIH experience of LPs. Patient experience of spinal anaesthesia and LP has previously been studied.[5 6] However, the experience of patients with IIH undergoing LP has not been evaluated. LPs are typically more technically challenging in the IIH population as over 90% of these patients are obese.[3]

The aim of this study was to assess the patient experience of diagnostic LPs in IIH. We aimed to disseminate this evidence to medical professionals to increase awareness of this potential morbidity of LP in patients with IIH. Furthermore, we aimed to use evidence from this study as a catalyst to drive improvements in patient care.

## MATERIAL AND METHODS
### Public and patient involvement
This research was initiated, designed and conducted by IIH UK, a charity that supports patients with IIH and carers. The charity agreed at a Trustee meeting to design a survey to investigate the magnitude of LP-related anxiety in response to the overwhelming messages from patients. When the first survey was performed, the trustees recognised that they would need help both in analysis of the data as well as asking additional questions. A further survey was therefore conducted. The clinical researchers at the University Hospitals Birmingham provided support with statistical analysis and critical review of the data.

Dissemination of the results was planned via physician and patient meetings, through medical and patient-led social media, and on the IIH UK patients' charity website.

## STUDY DESIGN
The cross-sectional study was conducted using an online survey. IIH UK sent a survey monkey questionnaire through social media outlets (Facebook, Twitter (@ IIHUK) and IIH UK charity website (www.iih.org.uk) and allowed a 2-month period from 1 April 2015 to 31 May 2015 for responses. Questionnaires were excluded if the respondents were under the age of 16 years or the survey was incomplete (missing key data fields) or uninterpretable. Anonymised data were analysed by the clinical team with input from the clinical research facility statistician (Peter Nightingale). As the charity board had already agreed with their members beforehand, and both surveys instructed the respondents that the information would be used to be published within the medical literature, no further ethical approval was required.

The questionnaire (see online supplementary document) detailed baseline demographic details (age, weight and height), and details of the LP (emergency vs planned procedure, hospital setting, number of attempts, whether went on to have an X-ray-guided procedure and seniority of doctor performing). Data on anxiety (for the LP and future LPs), pain experienced and extent of understanding of the procedure were also collected. Patients were asked to quantify responses using a Verbal Rating Score (VRS) 0–10 with 0 being the minimum and 10 the maximum score.

## Statistical analysis
Statistical analysis was performed using SPSS V.24 and GraphPad Prism V.7 for Windows (GraphPad Software, La Jolla, California, USA). Assessment of data for normality was performed for each analysis. Normally distributed data were reported using mean and SD, and non-normally distributed data were reported using median and IQR. For all comparisons of continuous variables, a non-parametric test was used due to non-normality of data distribution. For comparison of two medians Mann-Whitney U tests were used, while for comparison of multiple medians Kruskall-Wallis H tests were used. Spearman's rank-order correlation was used to analyse the correlation between how informed the patients were and how much pain they felt, as well as between body mass index (BMI) and how scared a patient was beforehand, how much pain they felt, and how anxious they felt about future LPs. For comparison of categorical variables $X^2$ tests were used. Values were considered statistically significant when p values were less than 0.05.

## RESULTS
### Demographics
There were 502 responders to the study, of which 463 were eligible for analysis. Eighteen responders did not complete the survey, 11 were under the age of 16 years and 10 gave incomplete answers or ambiguous information that could not be objectively interpreted (figure 1). The mean age was 33 years (SD 8.9), 98.5% were women (n=456), with a mean weight of 97.4 kg (SD 22.3), and a mean BMI of $36 \, \text{kg/m}^2$ (SD 8.3). The median number of LPs undergone since diagnosis was 4 (IQR 1–11), though 3.1% of patients (n=15) reported more than 50 LPs. When the number of LPs was adjusted to reflect length of disease, the median number of LPs per year since diagnosis was 1.3 (IQR 0.3–3.6) (table 1).

### Pain, anxiety and analgesia
The majority of patients indicated they were extremely scared about the imminent LP (median VRS 8, IQR 6–10), with 60% indicating a VRS ≥8 in relation to how scared they felt (figure 2A,B). Forty per cent of patients described severe pain during the LP (VRS ≥8) with a median pain score of 7 (VRS, IQR 5–8) (figure 2A,C). Additionally, the majority of patients felt they received insufficient pain relief (85%). Levels of anxiety about future LPs were high (median VRS 7, IQR 4–10), with 47% being extremely anxious (VRS ≥8) (figure 2A,D). There was no relationship found between the preprocedure anxiety levels and the subsequent recalled pain score of the LP.

### Setting of LP and preprocedural information
LPs were predominantly performed in the emergency setting (72%), as opposed to as an elective-planned

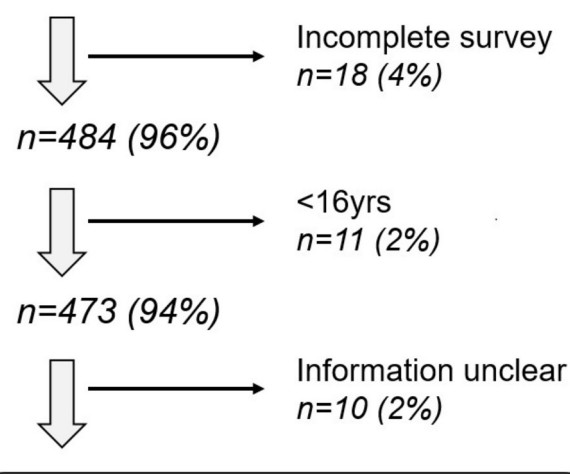

**Responders screened**
*n=502*
*(100%)*

→ Incomplete survey
*n=18 (4%)*

*n=484 (96%)*

→ <16yrs
*n=11 (2%)*

*n=473 (94%)*

→ Information unclear
*n=10 (2%)*

**Eligible**
*n=463*
*(92%)*

**Figure 1** Baseline characteristics of eligible responders.

procedure on day-case unit. Importantly, the LPs performed in the emergency setting were associated with significantly greater pain scores compared with elective procedures (VRS median 7, IQR 5–7 vs 6, IQR 4–8, respectively, p=0.012).

Only 37% of patients felt well informed about the LP pre-procedure (VRS ≥8); 27% felt poorly informed (VRS 0–3)and 7% did not feel they were informed at all (VRS 0). Higher LP pain scores (VRS) were significantly associated with patients being poorly informed (Spearman correlation, r=−0.32, p<0.001) (figure 3A). Patients felt better informed if they had an elective-planned LP

| Variable | No (%) n=463 |
|---|---|
| Age, years, mean (SD) | 32.9 (8.9) |
| Female sex | 456 (98.5%) |
| Weight, kg, mean (SD) | 97.2 (22.5) |
| BMI, mean (SD) | 36.0 (8.3) |
| LPs since diagnosis, median (IQR) | 4 (1–11) |
| LPs per year since diagnosis, median (IQR) | 1 (1–4) |

**Table 1** Baseline characteristics of eligible responders

BMI, body mass index; LP, lumbar puncture.

compared with a procedure in the emergency setting (median 7, IQR 5–10 vs median 6, IQR 5–10, p=0.011).

### Difficulty of LP and need for X-ray-guided procedure

Forty-seven per cent of patients had two or more doctors attempt their LP (median 1, IQR 1–2) while 45% had greater than three attempts (number of times needle inserted) before success. 10.7% went on to have an X-ray-guided procedure due to failure of the initial LP, and the BMI was significantly higher in this group (mean kg/$m^2$ vs 440.3 vs 35.5, p=0.001) (figure 3B). There was only a weak correlation between BMI and how scared a patient was beforehand, how much pain they felt, and how anxious they felt about future LPs (Spearman's r=0.17, 0.17, 0.17 respectively, p<0.001 for all). Compared with those that had normal LPs, the patients having X-ray-guided procedures felt less informed (VRS median 3 vs 6, p=0.002), suffered more pain (VRS median 8 vs 7, p=0.004) and were more anxious about future LPs (VRS median 9 vs 7, p=0.003).

### Grade of doctor performing LP

Table 2 shows the number of LP attempts by grade of doctor performing the LP. Patients felt more informed when the LP was performed by a specialist registrar (SpR) compared with a junior doctor (median VRS 7 vs 5, p=0.001) or a consultant compared with a junior doctor (median VRS 8 vs 5, p<0.001), though there was no significant difference in the pain scores reported. They also suffered from less severe post-LP headaches (SpR vs junior median VRS 7 vs 8, p<0.001, consultant vs junior median VRS 6.5 vs 8, p=0.003) (figure 3C), and length of post-LP headache (SpR vs junior median days 3 vs 6, p=0.02, consultant vs junior median days 4 vs 6, p=0.9) (figure 3D).

### DISCUSSION

This is the first study, to the best of our knowledge, to document the patient experience of diagnostic LP in IIH. It has shown that a number of patients are recalling significant pain and anxiety. This morbidity is associated with inadequate preprocedural information, the environment the LP is performed in (emergency setting being associated with increased pain), and the seniority of the doctor performing the LP.

Anaesthetists have long recognised the importance of the patient experience of spinal anaesthesia as an outcome measure and an indicator of quality of care.[5–7] This is reflected in the high satisfaction levels patients report with the procedure (96%–97%), which is in stark contrast to the feedback here. The differences between the anaesthetic population and the patient with IIH group may be related to the procedure being technically more challenging due to the patient's high BMI, the procedure happening as an emergency and some having multiple LPs during the course of their IIH. It may also be due to anaesthetists having better technical skills due to

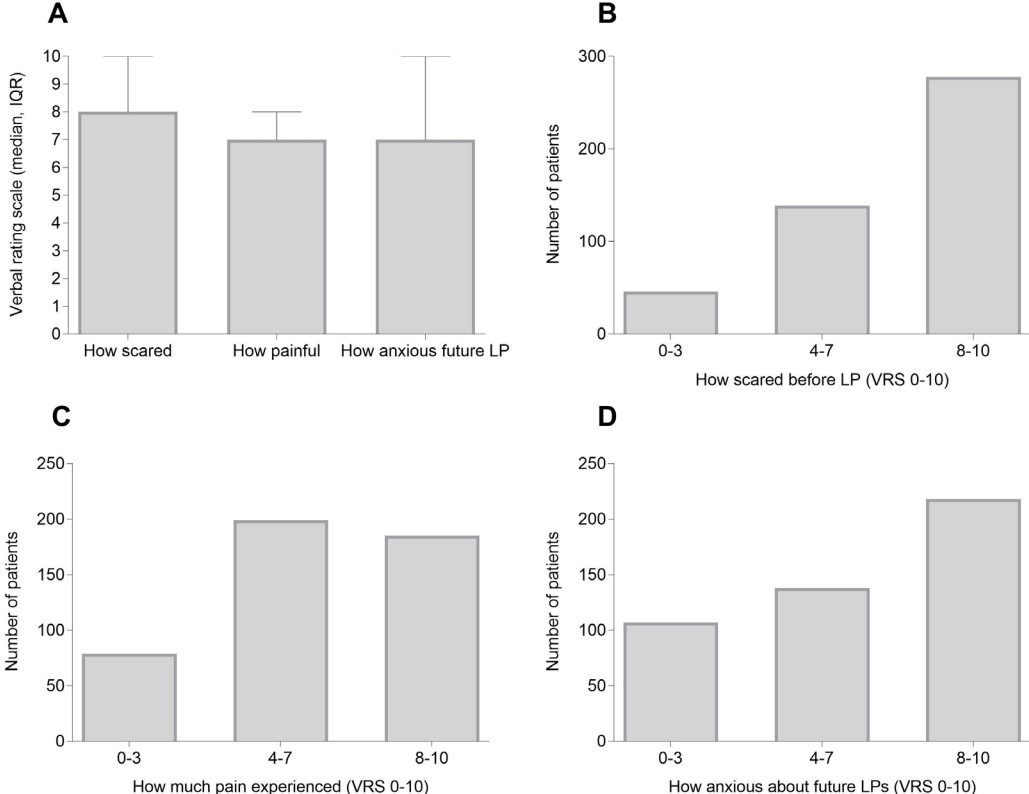

**Figure 2** Patients' expectations and experience of LP. (A) Median VRS (0–10, IQR) for how scared patient was before LP, how painful the LP was and how anxious they were about future LPs. (B) Number of patients who were mildly (0–3), moderately[4–7] or very scared[8–10] before having an LP. (C) Number of patients who experienced mild (VRS 0–3), moderate (VRS 4–7) or severe (VRS 8–10) pain during the LP. (D) Number of patients who were mildly (VRS 0–3), moderately (VRS 4–7) or very anxious (VRS 8–10) about future LPs. (VRS 0=minimum and 10 maximal score). LP, lumbar puncture; VRS, Verbal Rating Score.

performing the procedure more often than the doctors (often non-neurologists) performing the LPs in the emergency setting, in addition to more closely supervised and rigorous training.

This was a large sample size study (463 responders) where patients could respond anonymously, thus increasing the likelihood of honest reporting of their subjective experiences of the procedure. This cohort reports the LP experience as negative with 40% of patients experiencing severe pain (VRS ≥8) during the procedure, 85% saying they did not receive adequate analgesia and 47% stating they were extremely anxious (VRS ≥8) about future LPs.

The majority of the group did not feel they received adequate preprocedural information, with 63% not feeling well informed (VRS <8), and 7% saying that they were not informed at all (VRS=0). Patients who were less informed experienced more pain during the procedure. Although all patients will have undergone a consent process in the UK, these data highlight the variable quality of the information disseminated by the physician to the patient. Current practices for informing patients about LP are likely to be highly variable across the UK. This study highlights a key area where simple changes in clinical practice to ensure all patients are provided with detailed preprocedure information could facilitate improved patient care.

Environment also had a bearing on the patient experience with 72% of LPs being performed in the emergency setting; this was associated with the patient feeling less informed, and reporting significantly higher pain scores, compared with an elective procedure on a day-case unit. The high proportion of the respondents who had an LP performed as part of an emergency admission in this study likely reflects the UK healthcare services and clinical practice where patients with a flare up in IIH symptoms are typically initially seen in the accident and emergency department for initial evaluation and often have a LP as part of their evaluation here, or on the acute medical unit. As the study was not designed to determine the clinical indications for the LP in each case, no further inference can be made here. Typically, LPs performed in the emergency setting may be conducted by junior physicians, with less experience in conducting LPs than a specialty-trained neurologist or anaesthetist. This may be a factor contributing to the poorer outcomes from LPs performed in the emergency setting.

Optimisation of the environment for the patient undergoing LP could therefore positively affect their outcome. The day-case environment may provide access to doctors adequately trained in performing LPs as well as a less time-pressurised environment. Time to reflect on the procedure and read preprocedural information

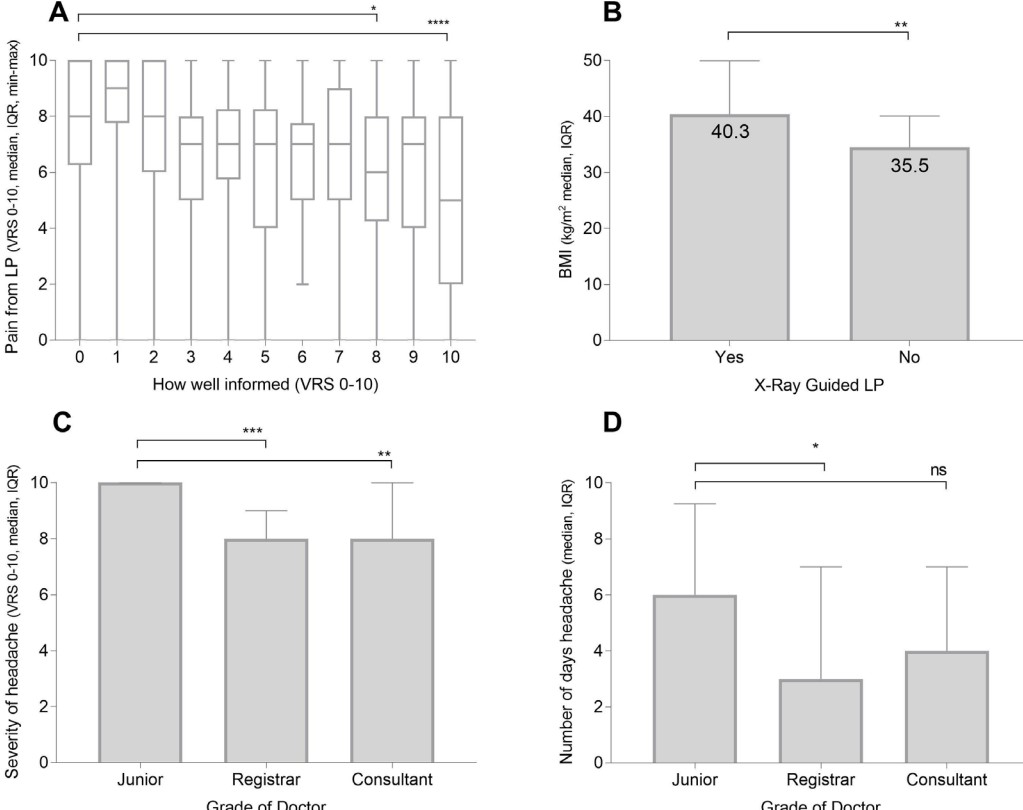

**Figure 3** X-ray-guided LPs, and relationship of preprocedural information and grade of doctor performing LP to patient experience. (A) For all patients surveyed, association between how well-informed patient was before LP, and how painful LP was (median VRS, IQR, minimum–maximum). (B) BMI (median, IQR) and association with whether patient had X-ray-guided LP. (C) Grade of doctor performing LP and duration of post-LP headache (days, median, IQR). (D) Grade of doctor performing LP and severity of post-LP headache. ns not significant, p>0.05, *p≤0.05, **p≤0.01, ***p≤0.001, ****p ≤ 0.0001 (VRS 0=minimum and 10=maximal score). BMI, body mass index; LP, lumbar puncture; VRS, Verbal Rating Score.

may help improve the overall experience. Although diverting patients with IIH away from the emergency setting would likely benefit the majority, it may not be practical for a minority of papilloedema cases where there is progressive or rapid loss of visual function, and the need for acute diagnosis.

The study also suggests that there is also scope for improving our technical skills in LP, as 85% of our cohort stated that they did not receive adequate analgesia, with 45% undergoing greater than three attempts (defined as the needle being fully withdrawn between attempts), and 47%

having two or more doctors attempt the LP. When the LP was performed by a doctor more junior than registrar (30% of the time), the patients felt less informed, and reported more severe and longer lasting post-LP headaches. We acknowledge that the grade of doctor performing the LP may not be accurately recalled by the patient and maybe more of a reflection of the patient's confidence in the doctor. However, diverting LPs into the day-case setting would provide an opportunity where the junior doctors could be appropriately supervised and trained, which has been shown to increase their ability to perform the procedure.[8]

| Table 2 | Number of lumbar puncture (LP) attempts by grade of doctor | | | | |
|---|---|---|---|---|---|
| | Grade of doctor (% of total patients (n=463)) | | | | |
| No of LP attempts | Unknown | Junior | Registrar | Consultant | Total |
| 1–3 | 13.0 | 9.1 | 15.1 | 11.0 | 48.2 |
| 4–6 | 4.8 | 5.2 | 6.3 | 3.9 | 20.1 |
| 7–9 | 2.6 | 1.9 | 3.0 | 1.7 | 9.3 |
| 10–14 | 0.6 | 1.7 | 1.3 | 0.4 | 4.1 |
| 15–19 | 0.4 | 1.1 | 0.4 | 0.0 | 1.9 |
| 20+ | 0.6 | 1.1 | 0.4 | 0.2 | 2.4 |
| Unknown | 4.5 | 1.1 | 3.2 | 5.2 | 14.0 |
| Total | 26.6 | 21.2 | 29.8 | 22.5 | 100 |

## Box 1 Recommendations for improving patient experience of diagnostic lumbar puncture (LP) in idiopathic intracranial hypertension

► Providing enhanced preprocedural information.
► Where possible, diverting emergency department LPs to elective procedures on dedicated day-case units.
► Simulation training for doctors and specialist nurses to develop appropriate technical (including ultrasound guidance) and human factor skills (such as communication, empathy and leadership) for performing LPs in a technically difficult patient cohort.
► Implementing widespread patient-reported outcome measures for LPs to guide the need for service improvements and training needs.

In this cohort, 10.7% of patients reported having an X-ray-guided procedure due to failure of the initial LP, with the BMI being significantly higher in this group (mean $40.3\,kg/m^2$ vs $35.5\,kg/m^2$, p=0.001). This finding is in keeping with a recent study which showed a strong correlation between BMI and procedure failure, with half of the failed LPs occurring in patients with a BMI greater than $35\,kg/m^2$.[9] The inability to palpate landmarks in obese patients is likely to be a significant driver of this correlation.[10] The X-ray-guided LP group felt less informed, reported more pain and were more likely to feel anxious about future LPs; findings most likely due to the number of failed attempts before the X-ray-guided procedure. The growing evidence base for use of ultrasound guidance, particularly in patients with a higher BMI and absence of landmarks,[11] suggests that its use in the cohort of patient with IIH may increase the success rate of the initial LP and decrease the number that require X-ray-guided procedures.

For clinical care a positive experience of a diagnostic LP will positively impact on the patient's future engagement with healthcare services, while in IIH research LP experience affects recruitment to clinical trials[12]; it is therefore critical that clinicians optimise patient care.

### Limitations

Given the self-report nature of this study, the results are likely susceptible to recall bias. Interpretation of some of the study questions is problematic, for example, for questions such as the number of attempts for a LP and the seniority of the doctor performing the procedure, the respondents may not accurately know the answer.

### CONCLUSION

There has been a growing consensus in recent years that if healthcare services are to better deliver patient-centred care then research needs to be more reflective of patients' needs and concerns.[13 14] This study was commissioned by the IIH patient group and is the first to document the patient experience of diagnostic LPs in IIH. It documents experiences of significant pain and anxiety associated with both inadequate preprocedural information and the setting the LP is performed in. The study suggests a number of practical steps that may improve the patient experience of LPs box 1.

#### Author affiliations
[1]Metabolic Neurology, Institute of Metabolism and Systems Research, College of Medical and Dental Sciences, University of Birmingham, Birmingham, UK
[2]Department of Neurology, University Hospitals Birmingham NHS Foundation Trust, Birmingham, UK
[3]Birmingham Neuro-Ophthalmology Unit, Ophthalmology Department, University Hospitals Birmingham NHS Trust, Queen Elizabeth Hospital Birmingham, Birmingham, UK
[4]Idiopathic Intracranial Hypertension UK, Patient Charity, UK
[5]Centre for Endocrinology, Diabetes and Metabolism, Birmingham Health Partners, Birmingham, UK

**Acknowledgements** With thanks to Peter Nightingale (PN) (Senior Statistician, University of Birmingham), for his advice on question framing for quantitative analysis of responses, as well as in planning and interpretation of statistical analysis, and to the IIH UK charity, as well as the IIH patient group without which this study would not have been possible.

**Contributors** WJS: compilation of the survey results, statistical analysis, interpretation of the results and drafting the manuscript. SPM: conception and design of the survey, critical review of the manuscript. TW: compilation of the survey results and statistical analysis. SD and SW: conception and design of the survey. HB: statistical analysis and interpretation of the results. KM: interpretation of the results and drafting of the manuscript. AY: interpretation of the results and drafting of the manuscript. AJS: conception and design of the survey, interpretation of the results and critical review of the manuscript.

**Funding** AJS is funded by an NIHR Clinician Scientist Fellowship (NIHR-CS-011-028) and by the Medical Research Council, UK (MR/K015184/1). This work was supported by the Midlands Neuroscience Teaching and Research Fund.

**Disclaimer** The views expressed are those of the author(s) and not necessarily those of the NHS, the NIHR or the Department of Health.

**Competing interests** None declared.

**Patient consent** Not required.

**Provenance and peer review** Not commissioned; externally peer reviewed.

**Data sharing statement** We are happy to share all data, including raw data if requested.

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
