## [Reviewer comments · BMJ Open]

ARTICLE DETAILS

TITLE (PROVISIONAL)	Characterising the patient experience of diagnostic lumbar puncture in Idiopathic Intracranial Hypertension: a cross-sectional online survey.
AUTHORS	Scotton, William; Mollan, Susan; Walters, Thomas; Doughty, Sandra; Botfield, Hannah; Markey, Keira; Yiangou, Andreas; Williamson, Shelley; Sinclair, AJ

VERSION 1 – REVIEW

REVIEWER	Shanthi Viswanathan Department of Neurology Kuala Lumpur Hospital Kuala Lumpur Malaysia
REVIEW RETURNED	26-Dec-2017

GENERAL COMMENTS	This is a very well written manuscript that identifies a "real world problem" of the quality of life issues of lumbar punctures in a large cohort of patients with Idiopathic intracranial hypertension. It first looks at identifying the evidence for the challenges of LP in a cohort of BIH patients, then for the contributing factors for these challenges from a clinical, technical and management point of view. And finally, very clearly outlines certain avenues for improvement. There are just a few improvements suggested section by section: Title: Is clear and defines the study/manuscript well ie patient's experience though it also looks at the factors contributing to their outcome/experience. Abstract: Need to expand VRS and IQR for the readers in the abstract In conclusion part of the abstract ie strengths and limitations line 42 to 43: spelling mistake with regard to "we scan.." which should be we can It would be nice to mention in the abstract as well the no that needed to have Xray guided LP's but the stress of no of words allowed negates it, but maybe the authors could consider this. The Introduction, design, methodology on patient recruitment was well discussed though it would be useful to understand the ethics requirements and the ethical approval for this study that was done in collaboration with the IIH UK patient group in order to strengthen this nice manuscript for the benefit of the requirements of the publication.
---

	Further, in the results was there any attempt to look at the parameter of degree of obesity (BMI) and correlation with the technical difficulties of LP rather than expressing it as a percentage as well and the anxiety levels. Discussion: the discussion was handled well and the references were well quoted. The tables were interesting and accurate, adding much to the clarity of the manuscript. However, the grade of expertise involved and the response of the patient and the place where LP was done which was mentioned in the text and abstract was not reflected in the tables as neither was the number of attempts which is important. Table 3 is made up of part 3A where the cohort who had higher BMI had xray guided LP but does not explain if patients in table 3B are made up of the Xray guided LP cohort who were less well informed or as mentioned in the text this was the overall group who felt more pain as they were less informed so correction of the headings would help and modification of the table may help. Strobe check lists and questionnaire was appropriate, and consent was through the society. Overall, this manuscript is novel, with much new knowledge and the information is important as it identifies a real world challenge of patient reported outcomes to lumbar puncture in IIH as well as the factors associated with it and identifies important possible improvements for the management of this group of patients in the future. It adds important information to IIH management that has not been explored before and new vital avenues in patient IIH care.
--	--

REVIEWER	Charly Gaul Migraene- and Headache Clinic Koenigstein, Germany
REVIEW RETURNED	28-Jan-2018

GENERAL COMMENTS	The authors present the results of a retrospective survey about experiences with diagnostic lumbar puncture in patients suffering from idiopathic intracranial hypertension. Due to obesity lumbar puncture is difficult in part of the patients and a correlation between weight was found. Additional factors influencing the patient satisfaction, the pain associated with the procedure, and the amount of anxiety are the fact who was performing the puncture (senior physician vs. a junior), performing the lumbar puncture in an emergency department (vs. elective) and amount of information about the procedure which the patient get before. However, some points are questionable: 72% of the lumbar punctures were performed in the emergency setting which is surprisingly to the reviewer. Vision lost is a dangerous symptom in patients suffering from IIH but not in 72% of the patients. Mostly they claim headaches and vision loss is usually not so dramatic that there is a need to perform the lumbar puncture as a emergency treatment. This might be a consequence of the UK health care setting and needs more information on the specific setting. In addition, the reviewer suspect that standardized patient information letters for informed consent into the lumbar puncture are in use, so more than a minimum of information is given to the patient. Lumbar puncture is an invasive procedure so an informative conversation should be standard. The authors are requested to
--

	comment on the specific habits in UK health care system. In the discussion the authors wrote “It may also be due to anaesthetists having better technical skills due to performing the procedure more often...”. The reviewer is surprised about this statement because lumbar puncture is daily routine for neurologists. Please comment if the specific situation in UK is different. In addition, there is no question about the area of expertise (anaesthetists vs neurologist) included in the questionnaire.
--	--

VERSION 1 – AUTHOR RESPONSE

We would like to thank the reviewers for their learned comments and help in shaping this manuscript.

Reviewer: 1

Reviewer Name: Shanthi Viswanathan

Institution and Country: Department of Neurology, Kuala Lumpur Hospital, Kuala Lumpur, Malaysia

Please state any competing interests or state ‘None declared’: None to declare

Please leave your comments for the authors below

This is a very well written manuscript that identifies a "real world problem" of the quality of life issues of lumbar punctures in a large cohort of patients with Idiopathic intracranial hypertension. It first looks at identifying the evidence for the challenges of LP in a cohort of BIH patients, then for the contributing factors for these challenges from a clinical, technical and management point of view. And finally, very clearly outlines certain avenues for improvement.

There are just a few improvements suggested section by section:

Title: Is clear and defines the study/manuscript well ie patient's experience though it also looks at the factors contributing to their outcome/experience.

Changed to: Negative patient experience of lumbar puncture in Idiopathic Intracranial Hypertension: a cross-sectional online survey.

Abstract: Need to expand VRS and IQR for the readers in the abstract

Additional clarification inserted into the Abstract (line 64-66): “Patients were asked to quantify responses using a verbal rating score (VRS) 0-10 with 0 being the minimum and 10 the maximum score.”

In conclusion part of the abstract ie strengths and limitations line 42 to 43: spelling mistake with regard to "we scan.." which should be we can

Updated, with thanks. (change between 87-88)

It would be nice to mention in the abstract as well the no that needed to have Xray guided LP's but the stress of no of words allowed negates it, but maybe the authors could consider this.

I have put this into abstract though obviously adds a few words to the abstract. Hopefully the Editor will allow this, if not we will take out. (Lines 74-77): “10.7% went on to have an X-Ray guided procedure due to failure of the initial LP, and the BMI was significantly higher in this group (mean kg/m² 40.3 vs. 35.5, p=0.001).”

The Introduction, design, methodology on patient recruitment was well discussed though it would be useful to understand the ethics requirements and the ethical approval for this study that was done in collaboration with the IIH UK patient group in order to strengthen this nice manuscript for the benefit of the requirements of the publication.

We agree we have added:

Line 113-119: “This research was initiated, designed and conducted by IIH UK, a charity who supports IIH patients and carers. The charity agreed at a Trustee meeting to design a survey to investigate the magnitude of lumbar puncture related anxiety in response to their overwhelming messages from patients. When the first survey was performed, the Trustees recognised that they would need help in analysis of the data and ask additional questions so a further survey was conducted. The clinical researchers at the University Hospitals Birmingham provided support with statistical analysis and critical review of the data.”

And Line 129-131: “As the charity board had already agreed with their members beforehand, and both surveys instructed the respondents that the information would be used to be published within the medical literature, no further ethical approval was required”.

Further, in the results was there any attempt to look at the parameter of degree of obesity (BMI) and correlation with the technical difficulties of LP rather than expressing it as a percentage as well and the anxiety levels.

We were not able to do this analysis (correlating BMI with technical difficulty of LP) because this was a patient survey so there were no clinician-based questions in the survey allowing us to assign whether patients were a technically difficult LP pre-procedure or not (this is a topic being addressed in a current study we are doing and will write up soon).

We planned and analysed the statistics with our Statistician Peter Nightingale (University Hospital Birmingham, Clinical Research Facility, lead statistician). Correlation of LP attempts with BMI was not felt to be representative, as the data for LP attempts was categorical ((0, 0-3, 4-6, 7-9, 10-14, 15-19, 20+). which reduced the power and validity of this analysis. We chose instead to analyse the x-ray guided LP rate against BMI (using necessity for x-ray guided LP rate as a marker of technical difficulty). This is in the results section and Figure 3b.

We did correlate BMI against patient experience (anxiety, how scared and pain) and this demonstrated a weak correlation (how scared=0.17, 0.17=pain, anxiety=0.17, p<0.001 for all). The statistical analysis section has been updated (line 149-150, “, as well as between BMI and how scared a patient was beforehand, how much pain they felt, and how anxious they felt about future LPs”).

We have also added this data to the results section (line 188-191, “There was only a weak correlation between BMI and how scared a patient was beforehand, how much pain they felt, and how anxious they felt about future LPs (Spearman $r = 0.17, 0.17, 0.17$ respectively, $p < 0.001$ for all)”).

Discussion: the discussion was handled well and the references were well quoted.

The tables were interesting and accurate, adding much to the clarity of the manuscript. However, the grade of expertise involved and the response of the patient and the place where LP was done which was mentioned in the text and abstract was not reflected in the tables as neither was the number of attempts which is important.

1. We have updated figure 3 to include Figure 3C (Grade of Doctor performing LP and severity of post-LP headache) and Figure 3D (Grade of Doctor performing LP and length of post LP headache) to represent graphically the text in results section (line 200-202)
2. Location of the LP is found at 174-176: Setting of LP and pre-procedural information, "LPs were predominantly performed in the emergency setting (72%), as opposed to as an elective planned procedure on day-case unit." We do not feel this figure needs to be represented in a table as well, as would not add extra clarity.
3. We have enclosed a new table (Table 2 – between lines 365-366) to show the number of LP attempts by grade of the doctor performing the LP. The results section has been updated to reflect this (line 195 "Table 2 shows the number of LP attempts by grade of doctor performing the LP")

Table 3 is made up of part 3A where the cohort who had higher BMI had xray guided LP but does not explain if patients in table 3B are made up of the Xray guided LP cohort who were less well informed or as mentioned in the text this was the overall group who felt more pain as they were less informed so correction of the headings would help and modification of the table may help.

We have updated the heading for Figure 3 (line 359) to make this clearer ("For all patients surveyed, association between how well-informed patient was before LP...").

Strobe check lists and questionnaire was appropriate, and consent was through the society.

Many Thanks

Overall, this manuscript is novel, with much new knowledge and the information is important as

it identifies a real world challenge of patient reported outcomes to lumbar puncture in IIH as well as the factors associated with it and identifies important possible improvements for the management of this group of patients in the future. It adds important information to IIH management that has not been explored before and new vital avenues in patient IIH care.

Reviewer: 2

Reviewer Name: Charly Gaul

Institution and Country: Migraine- and Headache Clinic Koenigstein, Germany

Please state any competing interests or state 'None declared': None declared

Please leave your comments for the authors below

The authors present the results of a retrospective survey about experiences with diagnostic lumbar puncture in patients suffering from idiopathic intracranial hypertension. Due to obesity lumbar puncture is difficult in part of the patients and a correlation between weight was found. Additional factors influencing the patient satisfaction, the pain associated with the procedure, and the amount of anxiety are the fact who was performing the puncture (senior physician vs. a junior), performing the lumbar puncture in an emergency department (vs. elective) and amount of information about the procedure which the patient get before.

However, some points are questionable:

72% of the lumbar punctures were performed in the emergency setting which is surprisingly to the reviewer. Vision lost is a dangerous symptom in patients suffering from IIH but not in 72% of the patients. Mostly they claim headaches and vision loss is usually not so dramatic that there is a need to perform the lumbar puncture as a emergency treatment. This might be a consequence of the UK health care setting and needs more information on the specific setting.

We completely agree that 72% of IIH patients do not have fulminant IIH with imminent threat to vision requiring an LP as the emergency. Emergency in this study was defined as non-elective (i.e. admitted to hospital through A&E) as opposed to planned (sent an appointment by doctor and brought in electively either to the ward of a day-case unit). However, the finding that such a high % of LPs were performed during an emergency admission for patients presenting with what turns out to be IIH is in keeping with our experience as Neurologists in the UK and is borne out by this study. We were not able to analyse what the indications were for LPs from this study as this data was not collected. It would have been very useful to analyse this figure in more detail although these sub-group analyses would likely have lacked statistical power. We would imagine that a significant proportion underwent diagnostic LPs, due to presentation with papilloedema and headaches/TVOs/Diplopia and did not have an imminent threat to vision. We have added Line 235-240: "The high portion of the respondents who had an LP performed as part of an emergency admission in this study likely reflects the UK health care services and clinical practice where patients with a flare up in IIH symptoms are typically seen initially in the accident and emergency department for initial evaluation and often have a LP as part of their evaluation. As the study was not designed to determine the specific clinical indications for the LP in each case no further inference can be made here."

In addition, the reviewer suspect that standardized patient information letters for informed consent into the lumbar puncture are in use, so more than a minimum of information is given to the patient. Lumbar puncture is an invasive procedure so an informative conversation should be standard. The authors are requested to comment on the specific habits in UK health care system.

Again, we agree. For a patient to given valid informed consent, this requires that they are fully informed as regards to the procedure, the benefits and the risks. This study is not able to address the standard practice in different institutions in the UK, but best practice would certainly be to give the patient a standardised patient information leaflet, for them to read prior to coming to hospital for the procedure. However, the reality in the UK is that many A&E departments and acute medical units will not have written information on LPs. This study demonstrates that where good information is provided the

outcome for patients is enhanced. This this data suggests a change in clinical practice to always use information leaflets pre-LP. We are currently working to develop such a leaflet in collaboration with IIHUK. We also would like to perform a further study looking at variations in current practice in this area. We have deleted: "informed consent" at line 247, because it gives the impression informed consent was not obtained in the emergency room setting.

We have added (line 227-231): "Although all patients will have undergone a consent process in the UK, this data highlights the variable quality of the information disseminated by the physician to the patient. Current practices for informing patients about LP are likely to be highly variable across the UK. This study highlights a key area where simple changes in clinical practice to ensure all patients are provided with detailed pre-procedure information (see Box 1) could facilitate improved patient care."

We have amended the abstract (line 83-85): "Patient experience of LP may be improved through changing clinical practice to include universal detailed pre-procedural information."

In the discussion the authors wrote "It may also be due to anaesthetists having better technical skills due to performing the procedure more often...". The reviewer is surprised about this statement because lumbar puncture is daily routine for neurologists. Please comment if the specific situation in UK is different. In addition, there is no question about the area of expertise (anaesthetists vs neurologist) included in the questionnaire.

Again, we completely agree. Neurologists do perform LPs daily in the UK as well, and are probably on a par experience wise with anaesthetists, and though there is little evidence to support this, it is very much our personal opinion as well. However, the issue in the UK for many of these patients is that when they first present, it is often to the medical teams via A&E, especially in the smaller hospitals where there may not be Neurologists onsite. In this situation they often have LPs performed by junior medical staff who do not perform the procedure regularly, which is far from ideal for these patients as they are often technically challenging (due to high BMI). What this survey identifies is likely a reflection of this emergency pathway, though unfortunately there was no question in the survey asking which speciality the doctor performing the LP was in. As you highlighted above (and we are in complete agreement with) the majority of patients with IIH do not have fulminant IIH with imminent threat to vision, and it is these patients that would most benefit from having their diagnostic LP performed in a planned elective setting by a neurologist, or at a least a doctor that performs LPs regularly. We have updated the statement you highlighted to reflect that the comparison we are making is between anaesthetists and non-neurologists (line 216-217) – "than the doctors (often non-neurologists) performing the LPs in the emergency setting"

We have also added to line 246-249: "Typically, LP's performed in the accident and emergency department may be conducted by junior physicians, with less experience in conducting LP's than a speciality trained neurologist or anaesthetist. This may be a factor contributing to the poorer outcomes from emergency department LPs."

Authors must include a statement in the methods section of the manuscript under the sub-heading 'Patient and Public Involvement'.

This should provide a brief response to the following questions:

How was the development of the research question and outcome measures informed by patients' priorities, experience, and preferences?

Test inserted into the Patient and Public Involvement section of the methods, Line 113-119 (“This research was initiated, designed and conducted by IIH UK, a charity who supports IIH patients and carers. The charity agreed at a Trustee meeting to design a survey to investigate the magnitude of lumbar puncture related anxiety in response to their overwhelming messages from patients. When the first survey was performed, the Trustees recognised that they would need help in analysis of the data and ask addition questions so a further survey was conducted. The clinical researchers at the University Hospitals Birmingham provided support with statistical analysis and critical review of the data.”)

How did you involve patients in the design of this study?

Please see above

Were patients involved in the recruitment to and conduct of the study?

Please see above

How will the results be disseminated to study participants?

Text inserted into the Public and Patient Involvement section of the methods (Line 120-121; “Dissemination of the results was planned via physician and patient meetings, through medical and patient lead social media, and on the IIH UK patients’ charity website..”)

For randomised controlled trials, was the burden of the intervention assessed by patients themselves?

Not applicable to this study

Patient advisers should also be thanked in the contributorship statement/acknowledgements.

This has been done. Text inserted into acknowledgements section Line 35-36 (“.. and to the IIH UK as well as the IIH patient group without which this study would not have been possible.”)

VERSION 2 – REVIEW

REVIEWER	Shanthi Viswanathan Department of Neurology, Kuala Lumpur Hospital, Kuala Lumpur, Malaysia
REVIEW RETURNED	12-Mar-2018

GENERAL COMMENTS	I have gone through the revised version and feel that the authors have addressed all the concerns adequately and satisfactorily. Just some minor grammatical issues exist ie in abstract : Patients felt more informed when the LP was performed by (a) compared to a Junior Doctor and elsewhere in the manuscript which can be easily rectified when the authors read through prior to publication. With that, I feel this is an important study on patient reported outcomes that will impact on improving the quality of care for this group of patients in the UK.
--

REVIEWER	Charly Gaul Migraine and Headache Clinic Königstein, Germany
REVIEW RETURNED	18-Mar-2018

GENERAL COMMENTS	Thanks for revise according to all points of the reviewers.
---

VERSION 2 – AUTHOR RESPONSE

We would like to thank the reviewers for all of their input previously, and also the Editor for their learned comments and help in finalising this manuscript.

Editors comments to Author:

1. - Please revise the title of your manuscript to include the research question, study design and setting (location). This is the preferred format of the journal. We do not accept manuscripts with declarative titles.

We have revised the title to include the research question. It now reads “Characterising the patients experience of diagnostic lumbar puncture in Idiopathic Intracranial Hypertension: a cross-sectional online survey.” We hope this now meets BMI Open’s requirements.

2. ‘Strengths and limitations’ section of your manuscript (after the abstract):

- Please revise your abstract to remove the reference to Box 1 in the conclusion section. Please also remove the Box 1 prefix from the title.

- This section should contain five short bullet points, no longer than one sentence each, that relate specifically to the methods. For example #4 does not relate to the studies methods. Please revise this section accordingly.

Many thanks for highlighting these points. We have removed Box 1 from the conclusion section of the abstract and from the title of the “Strengths and Limitations section”. We have also removed the 4th bullet point as per your advice given that it does not directly relate to the study’s methods. We have also removed the term Box 2 from the text in line 231, as well as from the title in the Recommendations box (Line 290) to match the changes you have suggested with regards to Box 1. We hope this is satisfactory.

3. We were confused by the description of your study as 'retrospective' as you are prospectively recruiting participants in order to complete the online questionnaire.

For example, in the third bullet point of strengths and limitations section (after the abstract), as well as in the Limitations of your discussion, you describe your study as retrospective. Please see the following article for full details of retrospective versus prospective studies:

<http://www.bmj.com/content/302/6771/249>

Please revise these sections accordingly, we suggest "Given the self-report nature of this study.....". We take on board your point, and have updated the manuscript accordingly, replacing the word “retrospective” on lines 87-88 with “self-report”, and updating line 278 to read “Given the self-report nature of this study..”

Reviewer(s)' Comments to Author:

Reviewer: 1

Reviewer Name: Shanthi Viswanathan

Institution and Country: Department of Neurology, Kuala Lumpur Hospital, Kuala Lumpur, Malaysia

Please state any competing interests or state ‘None declared’: None to declare

Please leave your comments for the authors below

I have gone through the revised version and feel that the authors have addressed all the concerns adequately and satisfactorily. Just some minor grammatical issues exist ie in abstract : Patients felt more informed when the LP was performed by (a) compared to a Junior Doctor and elsewhere in the manuscript which can be easily rectified when the authors read through prior to publication. With that, I feel this is an important study on patient reported outcomes that will impact on improving the quality of care for this group of patients in the UK.

Many thanks for these comments. I have been through the manuscript and made minor grammatical changes where required.

11 Apr-2018

Dear Dr. Scotton,

Thank you for submitting your manuscript entitled "Characterising the patients experience of diagnostic lumbar puncture in Idiopathic Intracranial Hypertension: a cross-sectional online survey." (manuscript ID bmjopen-2017-020445.R2) to BMJ Open. This has been returned to you to address the following issues before it can be assigned to the Editor.

- Please provide box 1 caption in your main document.

Apologies not quite sure what you mean here? In the previous editors comments (See above) the request was "Please revise your abstract to remove the reference to Box 1 in the conclusion section. Please also remove the Box 1 prefix from the title."

When I looked through I saw that the (see BOX 1) reference in the conclusion had not been removed. I have now removed that which is hopefully what you were referring to? Please let me know if this is not the case, and please advise what it is you want changing. Many Thanks

- Please ensure that your MANUSCRIPT'S TITLE in your main document and Scholar One submission system are the same.

The scholar one submission title has been updated to match the MANUSCRIPT TITLE.

- Please ensure to have the same DATA SHARING STATEMENT both in your main document and in Scholar One.

The Scholar One DATA SHARING agreement has been updated to match the article.